# Reset First Resistive Switching in Ni_1−x_O Thin Films as Charge Transfer Insulator Deposited by Reactive RF Magnetron Sputtering

**DOI:** 10.3390/nano12132231

**Published:** 2022-06-29

**Authors:** Dae-woo Kim, Tae-ho Kim, Jae-yeon Kim, Hyun-chul Sohn

**Affiliations:** 1Department of Materials Science and Engineering, Yonsei University, Seoul 03722, Korea; daewoo.kim@yonsei.ac.kr (D.-w.K.); kkthh22@yonsei.ac.kr (T.-h.K.); jaeyeonkim@yonsei.ac.kr (J.-y.K.); 2Lam Research, Daesan-ro 288, Icheon-si 17336, Korea

**Keywords:** resistive random access memory, nickel oxide, nickel vacancy, reset-first resistive switching, oxygen partial pressure, conductivity, area dependence

## Abstract

Reset-first resistive random access memory (RRAM) devices were demonstrated for off-stoichiometric Ni_1−x_O thin films deposited using reactive sputtering with a high oxygen partial pressure. The Ni_1−x_O based RRAM devices exhibited both unipolar and bipolar resistive switching characteristics without an electroforming step. Auger electron spectroscopy showed nickel deficiency in the Ni_1−x_O films, and X-ray photoemission spectroscopy showed that the Ni^3+^ valence state in the Ni_1−x_O films increased with increasing oxygen partial pressure. Conductive atomic force microscopy showed that the conductivity of the Ni_1−x_O films increased with increasing oxygen partial pressure during deposition, possibly contributing to the reset-first switching of the Ni_1−x_O films.

## 1. Introduction

Resistive random access memory (RRAM) [1] has been widely studied as a candidate for next-generation non-volatile memory to overcome the limitations of conventional memories, such as flash memory and dynamic random access memory (DRAM). RRAM has a relatively low operation voltage with excellent program and erase speed [2]. In addition, the device could be fabricated in a simple metal–insulator–metal (MIM) [3] structure, enabling the high-density cell structure of a cross-bar array with 4F^2^ [4,5]. It was reported that numerous transition metal oxides, including Al_2_O_3_ [6,7], HfO_2_ [8,9,10], NiO_x_ [11,12,13,14], TiO_x_ [15,16], TaO_x_ [17,18], Nb_2_O_5_ [19,20], and Pr_1−x_Ca_x_MnO_3_ [21,22,23] show resistive switching (RS) characteristics. Moreover, various deposition techniques, such as sputtering [24,25,26,27,28], atomic layer deposition (ALD) [29] and pulsed laser deposition (PLD) [30] were used for the formation of such oxides. Notably, nickel oxide (NiO) film is one of the most widely studied oxides and is reported to have low operation power, a high on/off resistance ratio and is compatible with the CMOS fabrication process [31,32]. NiO has a rock salt structure composed of Ni^2+^ and O^2−^ and is a member of the strongly correlated 3d transition metal oxides that exhibit charge-transfer insulator behavior [33,34]. It is an insulating oxide with a wide bandgap (E_g_ ≈ 4.3 eV) due to the charge transfer gap caused by “Hubbard U” between the 2p and 3d states [34,35]. Therefore, the pristine state of NiO is typically the insulating state in RRAM [36,37]. The RS phenomenon in NiO has been mainly described as the formation and rupture of conductive filaments. This reversible resistance transition between the high-resistance state (HRS) and low-resistance state (LRS) is caused by applying electrical stress after an “electroforming” step [38]. It was suggested that oxygen atoms are migrated by the electric field, leaving oxygen vacancies (V_o_^2+^) at the vacated sites during the electroforming step; the adjacent Ni^2+^ atoms are changed to Ni^0^ to compensate for the charge state, resulting in a Ni filament [39,40,41]. The electroforming process degrades the chemical and physical properties of devices of MIM structure, affecting their reliability. The characteristics of RS uniformity also deteriorate because of non-uniform filament formation among MIM devices [42]. Moreover, electroforming requires additional high-voltage circuits, significantly reducing the device density. Therefore, research on devices that can be operated without an electroforming step is essential for realizing RS memories [43,44,45].

This study investigated the RS characteristics of off-stoichiometric Ni_1−x_O films for unipolar and bipolar RSs (URS and BRS, respectively). Particularly, it was demonstrated that nickel-deficient Ni_1−x_O films deposited under excessive oxygen partial pressure exhibit a reset-first RS without an electroforming step. An RRAM device with a reset-first RS could be an alternative to overcome the limitations of RRAM requiring an electro-forming step.

## 2. Experimental

MIM devices with Pt/NiO/Pt and Pt/NiO/TiN stacks were fabricated for electrical characterization. First, Ti/TiN adhesive layers with thicknesses of 10–50 nm were deposited onto SiO_2_ on a Si substrate using DC magnetron sputtering. Pt or TiN films were then deposited as bottom electrodes (BE). BE with various areas of 0.18~4.0 µm^2^ were formed to investigate the area-dependence of the electrical characteristics. After BE formation, off-stoichiometric Ni_1−x_O films with a thickness of 10 nm were deposited via reactive RF magnetron sputtering using a Ni target under various O_2_ partial pressures. During sputtering, the base and working pressures were less than 3 × 10^−3^ and 3 mTorr, respectively. During deposition, the RF power and temperature of the substrate were main-tained at 100 W and 400 °C, respectively. The fraction of the O_2_ partial pressure in the mixture of Ar and O_2_ varied from 10% to 50% for deposition. Finally, Pt top electrodes (TEs) with a thickness of 100 nm were formed using DC magnetron sputtering and a lift-off process. The electrical characteristics of the device were characterized using a Keysight B1500A analyzer at 21~23 °C. RS under DC bias was measured with a com-pliance current of 10 mA to avoid hard breakdown of the Ni_1-x_O films. The spatial distribution of conductivity in the pristine state was investigated using conductive atomic force microscopy (C-AFM) (Park Systems, XE-100) with a measurement bias of 3 V [46,47]. Grazing incidence X-ray diffraction (GI-XRD, Rigaku SmartLab), Auger electron spectroscopy (AES, PHI-700, ULVAC-PHI), and X-ray photoelectron spectroscopy (XPS, K-alpha, Thermo U. K.) analyses were conducted to investigate the crystallinity, composition, and valence states of Ni in the Ni_1−x_O films, respectively.

## 3. Results and Discussion

XRD analysis was conducted to investigate the crystallinity of Ni_1−x_O films. The XRD patterns of Ni_1−x_O films deposited under various O_2_ fractions are illustrated in Figure 1a. The peaks of NiO (111), NiO (200), NiO (220), and NiO (311) imply a polycrystalline structure [48]. NiO films, deposited with an O_2_ partial pressure fraction of 50% showed lower intensity with a more comprehensive full-width half maximum (FWHM), implying poorer crystallinity of NiO films. The XRD peak of the (111) plane shifted to lower diffraction with increasing O_2_ partial pressure, indicating an increase in the lattice constant with increasing O_2_ partial pressure, as shown in Figure 1b. The increase in the lattice constant could be ascribed to the increased strain effect as Ni vacancies increase with excessive O_2_ partial pressure [48,49,50]. Figure 1c shows the composition of Ni and O, estimated from AES analysis of the Ni_1−x_O films with various O_2_ partial pressures during deposition. The volume of Ni is gradually reduced with increasing O_2_ partial pressure, resulting in a Ni-deficient Ni_1−x_O film. The compositions of nickel oxide at 10% and 50% O_2_ partial pressures were estimated to be Ni_0.89_O and Ni_0.86_O, respectively.

Figure 2a shows the typical behavior of Pt/Ni_1−x_O/Pt stacks. The pristine Ni_1−x_O films deposited under an O_2_ partial pressure fraction of 10% offered an initial high resistivity [51] at an applied voltage of 1.77 V (1.4 MV/cm) on the TE. The film resistance changed from HRS to LRS during the forming step. The resistance state was changed back to HRS at 0.64 V (0.5 MV/cm) during the subsequent bias application, exhibiting reversible switching for the positive bias on TE. The difference between the forming voltage (V_form_) and set voltage (V_set_) was approximately 0.57 V. In contrast, pristine Ni_1−x_O films deposited under the 30% or 50% O_2_ ratio showed low resistance in the pristine state without the electroforming step and reset-first RS behavior, where the initial LRS state was changed to the HRS state, as shown in Figure 2b,c. While V_set_ is similar to that of Ni_1−x_O films for the O_2_ partial pressure fraction of 10%, the I_HRS_/I_LRS_ ratio decreased because of the overall high current level in the HRS state. In particular, the I_HRS_ between these oxygen partial pressure fractions showed that the 50% O_2_ ratio was 10 times higher than that of 30% O_2_. The I-V curves of TiN/Ni_1−x_O/Pt stacks are plotted in Figure 2d–f. The Ni_1−x_O film deposited under a 10% O_2_ partial pressure fraction show BRS [52] characteristics, as shown in Figure 2d. The pristine Ni_1−x_O film showed high resistivity, and the resistance state changed to LRS after the electroforming step with a negative bias on TE. The difference between V_form_ (−4.0 V) and V_set_ (−0.7 V) was approximately 3.3 V. On the contrary, the Ni_1−x_O film deposited under the 30% or 50% O_2_ partial pressure fraction showed reset-first BRS behavior for a positive voltage on the TE, as shown in Figure 2e,f.

Figure 3 shows the electric currents at 0.64 V of the Pt/Ni_1−x_O/TiN stacks in the LRS and HRS states, where Ni_1−x_O films were deposited at various O_2_ partial pressures. The mean values of I_HRS_ and I_LRS_ (red line) increased with the O_2_ ratio, suggesting that the Ni_1−x_O film conductivity depends on the O_2_ partial pressure, as shown in Figure 3a. The Ni_1−x_O films with a 10% O_2_ fraction required electroforming for resistive switching, but the Ni_1−x_O films with a 30% O_2_ fraction or higher showed reset-first RS behavior without electroforming. Figure 3b shows the electrical currents at 0.64 V in the LRS states, which has a similar tendency to the I_HRS_ with O_2_ partial pressure, but the slope was lower than that of the I_HRS_ state. The I_HRS_ and I_LRS_ showed the highest values for Ni_1−x_O films deposited under the 50% O_2_ partial pressure fraction.

To understand the nature of resistance switching, HRS and LRS resistances were measured from devices with BE of 0.18, 0.38, 2.00, and 3.69 μm^2^ at a bias of ±0.48 V. Figure 4a shows the area dependent resistance for BRS device with Ni_1−x_O films deposited by 10% O_2_ partial pressure fraction. The resistance of the HRS remained almost constant with decreasing geometric device area, while that of the LRS is almost independent of the device area. These area-independent characteristics imply that resistance switching through the device occurs in local regions, such as filament paths, rather than homogeneously distributed switching paths [53,54,55,56,57]. Meanwhile, the resistances of reset-first RS devices with Ni_1−x_O films deposited at 50% O_2_ partial pressure showed increased dependence on the device area, as shown in Figure 4b. Because the area dependence of the LRS for Ni_1−x_O films with 50% O_2_ partial pressure is close to that of Ni_1−x_O films with 10% O_2_ partial pressure, the nature of the RS is filamentary in the local area. The significant dependence of HRS on the Ni_1−x_O films with 50% O_2_ partial pressure is attributed to the reduced resistance of the Ni_1−x_O films, as shown in Figure 4b.

The DC, and AC endurance characteristics of the Ni_1−x_O device are shown in Appendix A. DC endurance in Appendix A was measured at a read voltage (V_read_) of ±0.25 V under a compliance current of 10 mA. The measured I_HRS_/I_LRS_ ratio is higher than 10^1^ even after 10^3^ cycles. Appendix A shows the AC endurance under pulse, which is measured with a set pulse of −0.95 V with 180 ns, a reset pulse of 1.2 V with 180 ns, and a V_read_ of 0.3 V conditions. The device has a uniform I_HRS_/I_LRS_ ratio even after 10^5^ cycles, which results in a stable RS property.

C-AFM measurements investigated the two-dimensional (2D) variation of the Ni_1−x_O film conductivity. Figure 5a illustrates the scheme of the C-AFM measurement. NiO/Pt and NiO/SiO_2_/Pt stacks were simultaneously formed on a sample to compare the differences during the current image mapping. Cross-sectional TEM images of the Ni_1−x_O films for C-AFM measurements are shown in Figure 5b. The sample-to-sample variation in the Ni_1−x_O thickness on the SiO_2_/Pt stacks was estimated to be within 15%. Therefore, we ignore the difference in conductivity due to thickness variation. Figure 5c–e show the current mapping images at a bias of 3 V from Ni_1−x_O films deposited under various O_2_ partial pressures. The left region of each mapping image represents the reference of the insulating SiO_2_ between the BEs and Ni_1−x_O films. The regions on the right represent the Ni_1−x_O films on the Pt BEs in their pristine state. Similar to the I-V characteristics of MIM devices, C-AFM showed an increased current through the Ni_1−x_O films with increasing O_2_ partial pressure. The conductive regions in the Ni_1−x_O film regions increased with increasing O_2_ partial pressure fraction, as shown in Figure 5d,e. In particular, the current distribution is relatively uniform in Ni_1−x_O film with a 50% O_2_ fraction. In contrast, films deposited under 10% O_2_ partial pressure fraction showed improved resistivity, as shown in Figure 5c.

The effect of the O_2_ partial pressure on the chemical bonding states in the Ni_1−x_O films is investigated through XPS analysis. Figure 6a–c show the Ni 2p_3/2_ peaks of Ni_1−x_O films deposited with various O_2_ partial pressures. Ni^0^, Ni^2+^ and Ni^3+^ states with binding energies of 852.5, 853.7, and 855.5 eV, respectively, are used for deconvolution of Ni 2p_3/2_ peaks [58,59].

The proportion of the Ni^3+^ state was estimated from the ratio of the Ni^3+^ peak area to the Ni^2+^ peak area. The Ni^3+^ valence state increased while the fraction of Ni^2+^ ions decreased with increasing O_2_ partial pressure (Figure 6a–c). The Ni^3+^ ratio in the film grown under 10% and 50% O_2_ partial pressure was estimated at 14.0% and 23.9%, respectively. Meanwhile, the Ni^0^ state at the 852.5 eV peak was not observed in our Ni 2p_2/3_ peak analysis, although it was considered a conductive path in previous studies [39,40,41]. Conventionally, Ni vacancies form in Ni-deficient NiO films with relatively excessive oxygen. It was reported that nickel deficiency could promote the further oxidation of Ni^2+^ ions, which can be expressed with Kröger–Vink notation, as follows [48,49]:(1)$$2Ni_{Ni}^{x}+\frac{1}{2}O_{2}\left(g\right)\;\to \;2Ni_{Ni}^{∙}+O_{o}^{x}+V_{Ni}^{”},$$
where $Ni_{Ni}^{x}$, $Ni_{Ni}^{∙}$, $O_{o}^{x}$, $V_{Ni}^{”}$ represent Ni^2+^, Ni^3+^, O^2−^, and ionized Ni vacancies, respectively. Ni^2+^ ions react with oxygen to generate ionized nickel vacancies and two Ni^3+^ ions, which affect the conductivity of the nickel oxide films. Therefore, it is shown that the increase in Ni^3+^ in Ni_1−x_O films is related to the increase in the current in the HRS state of MIM devices and C-AFM. It is expected that Ni deficiency in Ni_1−x_O films grown under high O_2_ partial pressure causes a high Ni^3+^ concentration, leading to a highly conductive state and possibly the reset-first RS behavior with reinforced localized conductive paths [39,60,61]. Further investigation is required to understand how excess Ni^3+^ ions produce the reset-first resistive switching behavior in Ni_1−x_O films.

## 4. Conclusions

In this study, the reset-first RS characteristics of off-stoichiometric Ni_1−x_O films were investigated. The RS behavior without the electroforming step was observed in the unipolar and bipolar off-stoichiometric Ni_1−x_O films. Ni^3+^ distribution contributes significantly to the conductivity of the pristine Ni_1−x_O films. The conductivity and Ni deficiency of pristine Ni_1−x_O films increased as the O_2_ partial pressure increased during a deposition as revealed by the C-AFM and AES results. Moreover, Ni^2+^ was further oxidized to Ni^3+^ as the O_2_ partial pressure increased, as revealed by the XPS results.

The Ni_2_O_3_ bonding by Ni^3+^ ions is related to the reset-first RS behavior without the electroforming step. This is advantageous in terms of device scale-down, making Ni_1−x_O films promising candidates for memory applications by overcoming the limitations of the electroforming step in RRAM.

## Figures and Tables

**Figure 1 nanomaterials-12-02231-f001:**
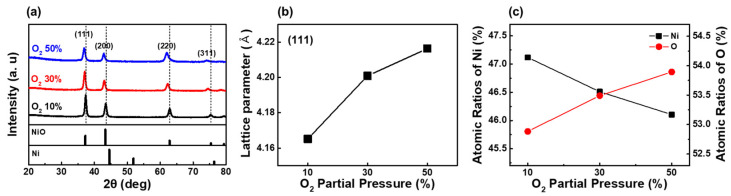
(**a**) XRD patterns of Ni_1−x_O films deposited with various oxygen partial pressures. (**b**) Lattice constant of Ni_1−x_O, estimated from (111) peak position, as a function of oxygen partial pressures. (**c**) Nickel and oxygen composition in Ni_1−x_O by AES.

**Figure 2 nanomaterials-12-02231-f002:**
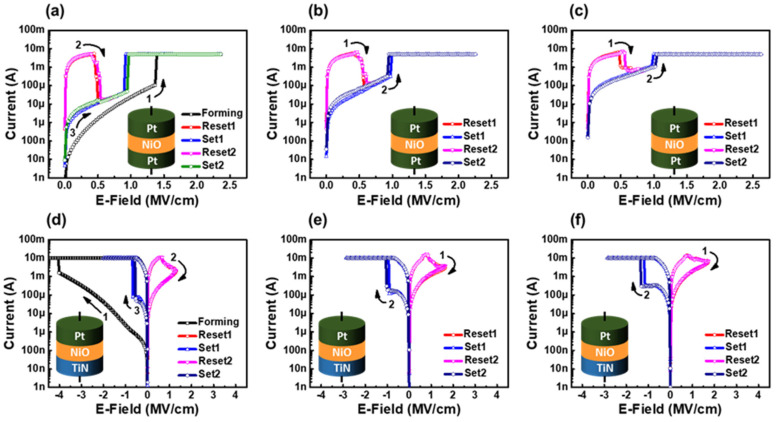
I−V characteristics of Ni_1−x_O devices with a bottom electrode of 2 × 2 μm^2^. URS characteristics of Ni_1−x_O films deposited with partial oxygen pressure of (**a**) 10%, (**b**) 30% and (**c**) 50%. BRS characteristics of Ni_1−x_O films deposited with oxygen partial pressure fraction of (**d**) 10%, (**e**) 30% and (**f**) 50%.

**Figure 3 nanomaterials-12-02231-f003:**
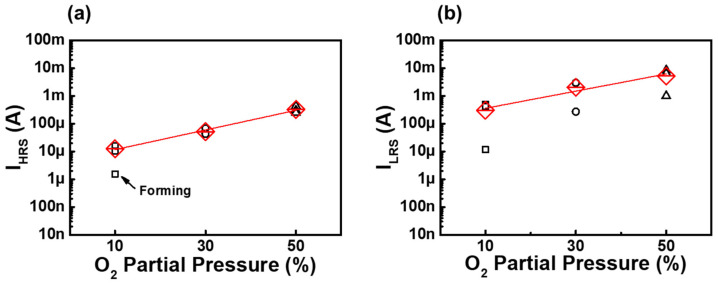
Influence of oxygen partial pressure on (**a**) I_HRS_ of Ni_1−x_O films and (**b**) I_LRS_ of Ni_1−x_O films.

**Figure 4 nanomaterials-12-02231-f004:**
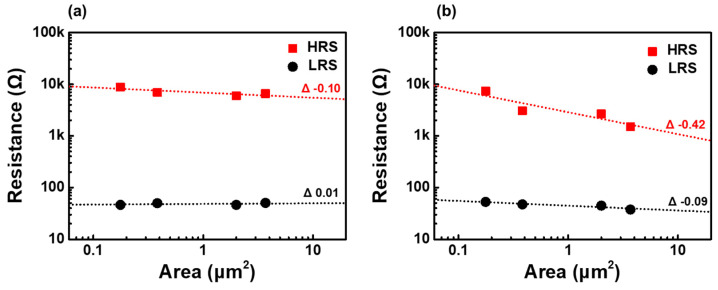
Area dependence of HRS and LRS resistances for Pt/Ni_1−x_O/TiN stacks (**a**) with Ni_1−x_O films, deposited with oxygen partial pressure fraction of 10%, with electroforming (**b**) with Ni_1−x_O films that are deposited with oxygen partial pressure fraction of 50%, with reset-first BRS without electroforming.

**Figure 5 nanomaterials-12-02231-f005:**
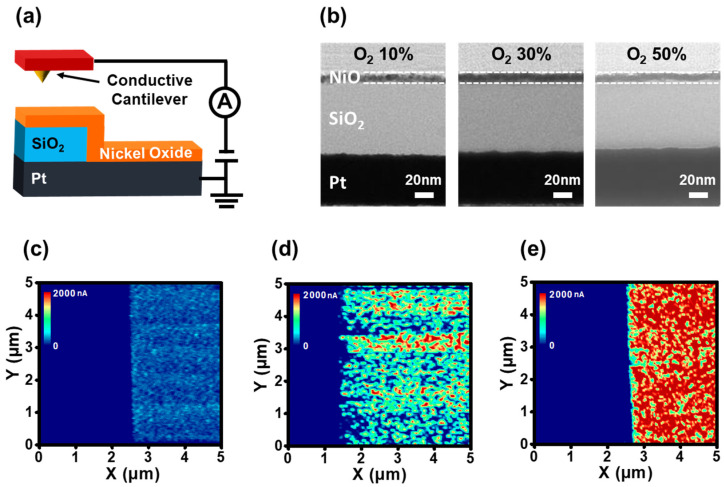
(**a**) Schematic diagram of the C-AFM measurement. (**b**) Cross-sectional TEM image of Ni_1−x_O films deposited at various oxygen partial pressure. C-AFM current mapping images of the pristine Ni_1−x_O films under oxygen partial pressure fraction of (**c**) 10%, (**d**) 30%, and (**e**) 50%.

**Figure 6 nanomaterials-12-02231-f006:**
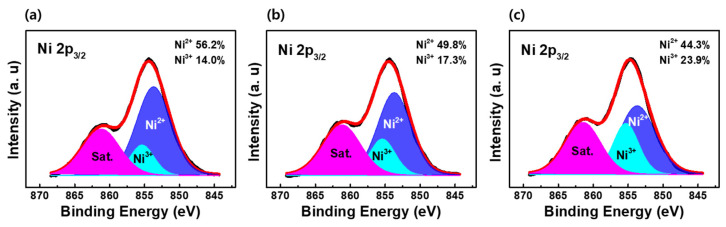
XPS peaks of Ni 2p_3/2_ of Ni_1−x_O films with oxygen partial pressure fraction of (**a**) 10% (**b**) 30% (**c**) 50%.

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
