# Peer review of "Reset First Resistive Switching in Ni1−xO Thin Films as Charge Transfer Insulator Deposited by Reactive RF Magnetron Sputtering"

_nanomaterials, 2022, doi:10.3390/nano12132231_

Round 1
Reviewer 1 Report
Referee Report
on paper “ Reset first resistive switching in Ni1-xO thin films as charge transfer insulator deposited by reactive RF magnetron sputtering “ (nanomaterials-1780050) by authors Daewoo Kim, Taeho Kim, Inwoo Kim and Hyunchul Sohn submitted to Nanomaterials
This is interesting and useful paper. It reports the preparation and investigation of the structure and electrical properties of the off-stoichiometric Ni1-xO thin films deposited using reactive sputtering with a high oxygen partial pressure. The Ni1-xO films exhibited both unipolar and bipolar resistive switching characteristics without an electroforming step. Auger electron spectroscopy showed nickel deficiency in the Ni1-xO films, and X-ray photoemission spectroscopy showed that the Ni3+ valence state in the Ni1-xO films increased with increasing oxygen partial pressure. Conductive atomic force microscopy showed that the conductivity of the Ni1-xO films increased with increasing oxygen partial pressure during deposition, possibly contributing to the reset-first switching of the Ni1-xO films. The obtained experimental results are reliable without any doubts. However, I have some questions and additions. I would like to note a few points to improve the paper before it can be published:
1. All the motivation should be deleted from the Abstract.
2. The authors should give examples in 1. Introduction of the formation of metal thin films by sputtering technique:
(1). A.I. Stognij, S.A. Sharko, A.I. Serokurova, S.V. Trukhanov, A.V. Trukhanov, L.V. Panina, V.A. Ketsko, V.P. Dyakonov, H. Szymczak, D.A. Vinnik, S.A. Gudkova, Preparation and investigation of the magnetoelectric properties in layered cermet structures, Ceram. Int. 45 (2019) 13030-13036. https://doi.org/10.1016/j.ceramint.2019.03.234.
(2). S.A. Sharko, A.I. Serokurova, N.N. Novitskii, V.A. Ketsko, M.N. Smirnova, A.H. Almuqrin, M.I. Sayyed, S.V. Trukhanov, A.V. Trukhanov, A new approach to the formation of nanosized gold and beryllium films by ion-beam sputtering deposition, Nanomaterials 12 (2022) 470. https://doi.org/10.3390/nano12030470.
3. I fully agree with the authors that: “Notably, nickel oxide (NiO) film is one of the most widely studied oxides and is reported to have low operation power, a high on/off resistance ratio and is compatible with the CMOS fabrication process.”. The authors should mention in 1. Introduction some results of the preparation and investigation of other substituted oxides promising for practcal applications:
(3). S.V. Trukhanov, A.V. Trukhanov, A.N. Vasil'ev, A. Maignan, H. Szymczak, Critical behavior of La0.825Sr0.175MnO2.912 anion-deficient manganite in the magnetic phase transition region, J. Exp. Theor. Phys. Lett. 85 (2007) 507-512. https://doi.org/10.1134/S0021364007100086.
(4). A.V. Trukhanov, S.V. Trukhanov, V.G. Kostishyn, L.V. Panina, V.V. Korovushkin, V.A. Turchenko, D.A. Vinnik, E.S. Yakovenko, V.V. Zagorodnii, V.L. Launetz, V.V. Oliynyk, T.I. Zubar, D.I. Tishkevich, E.L. Trukhanova, Correlation of the atomic structure, magnetic properties and microwave characteristics in substituted hexagonal ferrites, J. Magn. Magn. Mater. 462 (2018) 127-135. https://doi.org/10.1016/j.jmmm.2018.05.006.
4. The authors should give examples in 1. Introduction of the formation of thin films by other techniques:
(5). T. Zubar, V. Fedosyuk, D. Tishkevich, O. Kanafyev, K. Astapovich, A. Kozlovskiy, M. Zdorovets, D. Vinnik, S. Gudkova, E. Kaniukov, A.S.B. Sombra, D. Zhou, R.B. Jotania, C. Singh, S. Trukhanov, A. Trukhanov, The effect of heat treatment on the microstructure and mechanical properties of 2D nanostructured Au/NiFe system, Nanomaterials 10 (2020) 1077. https://doi.org/10.3390/nano10061077.
(6). T.I. Zubar, V.M. Fedosyuk, S.V. Trukhanov, D.I. Tishkevich, D. Michels, D. Lyakhov, A.V. Trukhanov, Method of surface energy investigation by lateral AFM: Application to control growth mechanism of nanostructured NiFe films, Sci. Rep. 10 (2020) 14411. https://doi.org/10.1038/s41598-020-71416-w.
5. The proposed 6 papers should be inserted in References.
The paper should be sent to me for the second analysis after the moderate revisions.
Reviewer 2 Report
This manuscript reported the influence of stoichiometry on the resistive switching properties of oxygen-rich Ni1-xO nanofilms deposited by RF magnetron sputtering. The films show both unipolar and bipolar resistive switching characteristics without extra electroforming. Characterizations on the structure and composition of the Ni1-xO thin films are given and well discussed, and the electrical behaviors and their evolution with different films are clearly shown. A revision is suggested before consideration of publication.
(1) Long-term cycling between HRS and LRS is suggested to evaluate the stability for the oxygen rich sample of Ni1-xO film.
(2) More discussion on the switching mechanism for the Ni1-xO thin film is suggested, especially the influence of defect and the reason of electroforming-free behavior.
(3) What are the I-V properties for samples prepared with oxygen partial pressure of 30%, as shown in Figure 2. Please provide some data and discuss.
Round 2
Reviewer 1 Report
Referee Report
on paper “ Reset first resistive switching in Ni1-xO thin films as charge transfer insulator deposited by reactive RF magnetron sputtering “ (nanomaterials-1780050-v2) by authors Daewoo Kim, Taeho Kim, Inwoo Kim and Hyunchul Sohn submitted to Nanomaterials
This paper has been well corrected and it should be published urgently.
Reviewer 2 Report
As the authors have revised the manuscript accordingly, it is acceptable for publication in the journal.